# Bioinformatic Assessment and Expression Profiles of the *AP2/ERF* Superfamily in the *Melastoma dodecandrum* Genome

**DOI:** 10.3390/ijms242216362

**Published:** 2023-11-15

**Authors:** Yuzhen Zhou, Ruiyue Zheng, Yukun Peng, Jiemin Chen, Xuanyi Zhu, Kai Xie, Qiuli Su, Ruiliu Huang, Suying Zhan, Donghui Peng, Kai Zhao, Zhong-Jian Liu

**Affiliations:** 1Ornamental Plant Germplasm Resources Innovation & Engineering Application Research Center, Key Laboratory of National Forestry and Grassland Administration for Orchid Conservation and Utilization, College of Landscape Architecture and Art, Fujian Agriculture and Forestry University, Fuzhou 350002, China; zhouyuzhencn@fafu.edu.cn (Y.Z.); fafuruiyue@163.com (R.Z.); pyk20001022@163.com (Y.P.); c2549539102@163.com (J.C.); zxy316377506@163.com (X.Z.); xiekai_0526@163.com (K.X.); sqljya@163.com (Q.S.); huangruiliu06@163.com (R.H.); jamaisvuzhan@163.com (S.Z.); fjpdh@fafu.edu.cn (D.P.); 2College of Life Sciences, Fujian Normal University, Fuzhou 350117, China

**Keywords:** *AP2/ERF* superfamily, transcription factor, *Melastoma dodecandrum*, expression profiles, crosstalk with IAA

## Abstract

AP2/ERF transcription factors play crucial roles in various biological activities, including plant growth, development, and responses to biotic and abiotic stressors. However, limited research has been conducted on the AP2/ERF genes of *Melastoma dodecandrum* for breeding of this potential fruit crop. Leveraging the recently published whole genome sequence, we conducted a comprehensive assessment of this superfamily and explored the expression patterns of *AP2/ERF* genes at a genome-wide level. A significant number of genes, totaling 218, were discovered to possess the AP2 domain sequence and displayed notable structural variations among five subfamilies. An uneven distribution of these genes was observed on 12 pseudochromosomes as the result of gene expansion facilitated by segmental duplications. Analysis of cis-acting elements within promoter sites and 87.6% miRNA splicing genes predicted their involvement in multiple hormone responses and abiotic stresses through transcriptional and post-transcriptional regulations. Transcriptome analysis combined with qRT-PCR results indicated that certain candidate genes are involved in tissue formation and the response to developmental changes induced by IAA hormones. Overall, our study provides valuable insights into the evolution of ERF genes in angiosperms and lays a solid foundation for future breeding investigations aimed at improving fruit quality and enhancing adaptation to barren land environments.

## 1. Introduction

Transcription factors (TFs) play pivotal roles in orchestrating gene expression networks in response to diverse environmental stresses and essential developmental processes through specific DNA sequence binding, thereby modulating chromatin structure and regulating transcription [1]. The *AP2/ERF* superfamily is characterized by the presence of the AP2/ERF domain, comprising approximately 60 to 70 amino acids, which plays a crucial role in DNA binding [2]. The AP2 (APETALA2) family of proteins contains two tandemly repeated AP2/ERF domains, while the ERF (ethylene response factors) family of proteins possesses a single AP2/ERF domain. In contrast, the RAV (related to ABI3/VP1) family proteins harbor a B3 DNA-binding domain that is conserved in other plant-specific transcription factors, in addition to the single *AP2/ERF* domain.

AP2 subfamily genes are implicated in the developmental processes of flowers and seeds [3,4]. In contrast, ERF subfamily genes specifically contribute to plant defense against biotic stressors and pathogens [5], while CBF/DREB subfamily genes primarily regulate plant responses to abiotic stressors such as osmotic and cold stress [6,7]. However, members of the RAV subfamily play diverse roles, including bud outgrowth and leaf senescence, as well as responses to pathogen infections and abiotic stresses [8,9]. A small subfamily, known as Soloist, possesses a single AP2 domain but exhibits highly divergent sequences compared to other *AP2/ERF* genes. These Soloist genes play pivotal roles in enhancing plant salt tolerance and response to pathogens [10,11,12]. Furthermore, multiple studies have studied the functional characterization of multiple members of the AP2 family in plant growth and development, including flower [13,14], seed [15,16], and leaf development [17]. TFs belonging to the ERF family are the important downstream regulators of the ethylene-mediated stress-response signaling pathways, which are involved in multiple processes, such as fruit ripening [18,19] and the resistance and tolerance to biotic and abiotic stresses, and crosstalk with other hormones [20,21].

With the advent of high-throughput sequencing, over 700 plant genomes have been released [22], greatly facilitating the identification and functional analysis of *AP2/ERF* genes. A plethora of *AP2/ERF* genes from diverse plant species have been identified and extensively utilized to enhance crop resilience against environmental stresses. This class of TFs has been comprehensively identified at a genome-wide level in well-established plant models such as Arabidopsis, rice [23], and tomato [24], and non-model plants including *Vitis vinifera* [25], *Liriodendron chinense* [26], Aquilaria sinensis [27], Pecan [28], and Populus [29,30]. To date, *AP2/ERF* gene mining in Melastomataceae, which contains about 166 genera and 4500 species, is rather limited [31]. Thus, further research on Melastomataceae plants and more accurate and detailed identification is required [32,33].

*M. dodecandrum* is a shrub that thrives in the natural progression of ecosystems, maintaining its evergreen nature [34]. This species can be found in habitats spanning from southern China to Vietnam. It exhibits remarkable adaptability to forest areas through its creeping stems, and also holds potential for agricultural exploitation due to its nutritious fruit. The fruit accumulates substantial quantities of tannins and flavonoids, known for their high antioxidant activity [35]. As a result, *M. dodecandrum* is recognized as an invaluable resource with significant economic, ornamental, and medicinal values. Recently, we have successfully compiled a genome of *M. dodecandrum* [31] at the chromosome level (299.81 Mb; 35,681 predicted genes; 12 pseudochromosomes). In this research, we identified *AP2/ERF* members in *M. dodecandrum* based on genome data and classified them into relative subfamilies combined with the *AP2/ERF* family in 20 other representative plant species, ranging from algae to angiosperms. We analyzed basic physical and chemical properties, gene structure, conserved motif, chromosomal distribution, cis-element, and microRNA binding sites. Through gene expression patterns in different tissues, we located the potential core functional members. This study represents a pioneering effort in genome-wide identification of *AP2/ERF* family genes of *M. dodecandrum*, providing valuable insights into the response and regulatory mechanisms of *AP2/ERF* genes.

## 2. Results

### 2.1. Genome-Wide Identification and Characterization Analysis of AP2/ERFs in Melastoma dodecandrum

Based on the *M. dodecandrum* genome data, 218 *AP2/ERF* genes were identified. A total of 176 *AtAP2/ERF*s and 218 *MedAP2/ERF*s were used to construct a phylogenetic tree to classify *MedAP2/ERF*s. Phylogenetic analyses divided the 218 *MedAP2/ERF* members into five subfamilies: AP2, DREB, ERF, Soloist, and RAV, based on the genome data (Figure 1). The classification results were confirmed with conserved domains (AP2 domains) and the classification of *AtAP2/ERF*s. Based on the phylogenetic tree, 24 *MedAP2/ERF*s were assigned to the cluster of the AP2 subfamily. A total of 188 *MedAP2/ERF*s were clustered in the ERF (109 TFs) and DREB (79 TFs) subfamilies. The Soloist subfamily included two members, and the RAV subfamily included four members in *M. dodecandrum* (Figure 1). In summary, we identified 24 AP2s, 79 DREBs, 109 ERFs, four RAVs, and two Soloists based on the *M. dodecandrum* genome.

The Gene ID and Gene name of the control are listed in Appendix A, and the physical and chemical properties and characteristics of MedAP2/ERF proteins can be found in Appendix A. According to the data for the MedAP2/ERF protein, the number of amino acids ranged from 76 to 1170, the molecular weight varied from 9.38 to 130.38 kDa, and theoretical pI changed from 4.6 to 11.45. The instability indexes were 37.07 to 95.1, aliphatic indexes were 32 to 87, the grand average of hydropathicity was less than 0 and ranged from −1.17 to −0.18; all of these were hydrophilic proteins.

Based on the phylogenetic tree clustering results for *A. thaliana* (Appendix A), we found that the number of Soloist subfamily, AP2 subfamily, and RAV subfamily members were 1, 7, and 3 less than those in *A. thaliana*, accounting for 33.33%, 22.58%, and 42.86% of the corresponding family members in *A. thaliana*, respectively. But the members of the DREB subfamily and ERF subfamily were 79 and 109, respectively, which were 36.21% and 41.56% more than those in *A. thaliana* (21 *AtDREBs* and 77 *AtERFs*). *M. dodecandrum* showed member expansion in the DREB subfamily and ERF subfamily.

### 2.2. Prediction of Secondary and Tertiary Structure of AP2/ERF TF Proteins

AP2/ERF protein’s secondary structure included an alpha helix, an extended strand, beta turn, and a random coil. This study used online software SOPMA to analyze the coding protein secondary structures of five AP2/ERF TFs by using the homology modeling method. Detailed prediction and analysis of the secondary and tertiary structure of each *AP2/ERF* TF subfamily in *M. dodecandrum* are shown in Figure 2A,B. Proteins were changed using different ratios of helix, turn, strand, and random coil (Figure 2C).

The protein sequences were submitted to SWISS-MODEL software online for tertiary structure prediction analysis (Figure 2B). The results showed that MedAP2-17, MedDREB-38, and MedERF-92 had similar ratios of alpha helix, extended strand, beta turn, and random coil. The random coil ratios of MedRAV-1 and Medsoloist-3 were significantly lower than that of other subfamily AP2/ERF proteins, and the extended strand ratio of MedRAV-1 was up to 23.81%. The tertiary structure of MedAP2/ERF protein was mainly composed of random coil, and the tertiary structures of MedAP2-17, MedERF-38, and MedDREB-92 were similar. There was a certain similarity between Soloist and AP2 subfamily of AP2/ERF TFs, while the tertiary structure of the RAV subfamily was quite different from that of other subfamily members.

### 2.3. Comparative Analysis of AP2/ERF Gene Subfamily Sizes across APG IV System

To elucidate the evolutionary trajectories of *AP2/ERF* families, we identified and analyzed *AP2/ERF* members across 24 plant species spanning algae, early land plants, and angiosperms, in addition to *M. dodecandrum* and *A. thaliana* (Figure 3A,B). Our analysis revealed that only AP2, ERF, and Soloist subfamilies were present in the three algal species (*V. carteri*, *C. braunii*, and *C. reinhardtii*). The DREB and RAV subfamilies first emerged in mosses (Figure 3A). While the DREB and ERF subfamilies were consistently represented by numerous members among diverse plant taxa. The RAV subfamily retained only a few members. The Soloist subfamily constituted a major proportion of algae, but comprised limited members in land plants. Based on the classical classification, 12 of the 13 eudicot species contained all *AP2/ERF* subfamilies, except for *E. songoricum* which lacked the RAV subfamily. The number of AP2/ERF members in these 13 eudicots ranged from 140 in *L. corniculatus* to 440 in *G. max* (Figure 3C). The number of members in the five subfamilies showed fluctuating trends across all species, especially for the AP2, DREB, and ERF subfamilies. *G. max* contained the highest number of AP2 and ERF subfamily members among all species, while *P. patens* had the most abundant DREB subfamily members. The counts of the five subfamilies in *M. dodecandrum* were at intermediate levels among the 24 species, consistent with the phylogenetic position of earthworms.

### 2.4. Chromosome Distribution Analysis of the AP2/ERF Genes

The number of MedAP2/ERF genes varies greatly between different chromosomes. Generally speaking, the longer the chromosome, the more *MedAP2/ERF* genes are distributed (Figure 4A). Chromosome 1 contained the most genes (40), while chromosome 4 contained the least *AP2/ERF* genes (only three genes), all of which were ERF subfamily members. The combination of genes in the chromosome distribution density (Figure 4B) found that the number of *MedAP2/ERF* genes on chromosome 4 was less than other chromosomes, implying that a relatively independent evolution of chromosome 4. 

The *MedERF* subfamily was the branch with the largest number of *AP2/ERF* superfamily members, and its genes were distributed on each chromosome. The *MedDREB* was distributed on all chromosomes except chromosome 4, and the *MedAP2* gene was distributed on all chromosomes except chromosome 4 and chromosome 9. Two *Medsoloist* genes were located on chromosomes 3 and 8, and four *MedRAV* genes were located on chromosomes 2, 3, 7, and 9. Studies have shown that *MedAP2/ERF* genes are unevenly distributed on chromosomes. MCScanX found three tandem repeats on chromosomes 3, 5, and chromosome 8 (Figure 4A), and 189 pairs of segmental duplications on all 12 chromosomes (Figure 4B). There were six *MedAP2/ERF* genes involved in tandem repeats, including three pairs of *MedERF* genes.

### 2.5. Gene Structure Contains All Subfamilies

There is a large variation in the structure of the AP2/ERF superfamily genes in *M. dodecandrum* (Figure 5, Figure 6 and Figure 7A; Appendix A). A total of 218 family members were found in all subfamilies, with AP2, DREB, and ERF subfamily members being the most numerous subfamilies. All AP2 subfamily genes had multiple introns ranging from three to ten. The soloist subfamily had five to nine introns, which was more similar to the AP2 subfamily members. All DREB subfamily members had no introns except *MedDREB-1*, *MedDREB-30,* and *MedDREB-51*, which had one intron, and *MedDREB-15*, which had three introns. All three RAV subfamily members had no introns except *MedRAV-4*, which had one intron. ERF subfamily members were more variable, with about 70% of the family members having no introns and the rest having one to twelve introns. *MedERF-61* was the most specific, containing 12 introns (Figure 7A).

Some subfamilies contained some specific conserved motifs (Figure 5 and Figure 6). Analysis of the conserved motifs in *M. dodecandrum* showed that the majority of AP2/ERF gene family members possessed both Motif 1 and Motif 2, and Motif 3. These three motifs were the major motifs that constituted the AP2/ERF gene family in *M. dodecandrum* (Figure 5). Meanwhile, conserved motifs of members of the same subfamily in *M. dodecandrum* exhibited a high degree of similarity and might perform similar biological functions. However, different subfamilies had different conserved motifs, which could be related to the regulatory functions of gene families. Among them, two soloist subfamily members contained only motif1, while the four RAV subfamily proteins contained only three motifs, motif1-3. In addition to MedERF-38, the AP2 subfamily had motif4 and motif6. The ERF subfamily had a unique motif8 and the DREB subfamily had motif5 and motif10.

### 2.6. Analysis of AP2/ERF Gene Duplication Events and Selection Pressure in M. dodecandrum

Gene replication is one of the most important mechanisms for plants to obtain and create new genes, and it is also the main driving force of genome evolution. Gene duplication usually includes tandem duplication, chromosomal segmental duplication, polyploidy, and single gene transposition-duplication. It was found that there were 179 pairs of duplicate genes in the AP2/ERF gene family, including *MedERF-52/MedERF-51*, *MedERF-82/MedERF-83,* and *MedERF-42/MedERF-43*. We found a total of three tandem repeats (Figure 4B), and 176 segmental duplications, such as *MedRAV-2/MedRAV-3*.

In order to further analyze the selection pressure of the AP2/ERF gene family, 179 pairs of homology genes were analyzed for Ka, Ks, and Ka/Ks ratio (Appendix A). The results showed that the Ka values of all gene pairs were less than the Ks value. All Ka/Ks values were less than one, indicating that the AP2/ERF gene family underwent a great purification selection pressure during the re-evolution of the *MedAP2/ERF* gene family to reduce the harmful mutations after fragment duplication, thereby maintaining its function. This reflects that they did not have much functional differentiation in the process of evolution and were highly conserved.

### 2.7. Target of Specific miRNAs for MedAP2/ERF Genes

In this study, we selected different members of the *MedAP2/ERF* gene family as candidate target gene sequences, and used the psRNATarget online software to predict the *miRNA*s of these different *MedAP2/ERF* genes. The results (Figure 7B; Appendix A) showed that each subfamily of *MedAP2/ERF*s had a different number of *miRNA* binding sites. Specifically, each gene in the 24 AP2 subfamilies had a different number of *miRNA* AP2 targets, totaling 187 targets. Among the 79 DREB subfamilies, 68 genes had 244 *miRNA* target sites, and 93 genes of the 109 DREB subfamilies had 349 *miRNA* target sites. All genes from the four RAV subfamilies had 21 *miRNA* target sites, while all genes from the two soloist subfamilies had five *miRNA* target sites, respectively (Appendix A). These findings suggest the existence of a complex network regulation system between *miRNA* and *MedAP2/ERFs*. Predicting miRNA targets that regulate genes in different AP2/ERF subfamilies helps provide clues for further studying the mutual regulatory relationship between *MedAP2/ERF*s and *miRNA*.

### 2.8. Cis-Acting Elements in the Promoter Regions of 218 MedAP2/ERF Genes

A total of 126 cis-elements were found (Figure 8), including 10 unnamed and 116 reported elements. They are involved in different biological functions (Appendix A). We selected 35 classical, previously reported homeostatic elements, which fall into 22 categories and classified them into two main functional categories, phytohormone responsive and abiotic or biotic stress elements. These 22 categories appeared 8901 times in the promoter regions of 218 *MedAP2/ERF* genes, among which MYB, ABRE, and STRE elements were the most abundant (Appendix A). In total, 100% of *MedAP2/ERF* genes contained MYB and STRE binding sites that appeared 1487 and 1196 times, accounting for 16.71% and 13.44% of abiotic and biotic stress-related cis-elements, respectively. ABRE was found 1457 times in 208 *MedAP2/ERF* genes, accounting for 16.37% of plant hormone response genes (Appendix A). In addition, JA response elements (CGTCA motif, TGACG motif), SA response elements (TCA element), GA response elements (GARE, P-box and TATCbox), ET response elements (ERE), and auxin response elements (TGA, AuxRR) were present in 199, 148, 137, 105, and 142 *MedAP2/ERF* genes, respectively (Appendix A).

### 2.9. Expression Patterns of AP2/ERF Gene Family Members in M. dodecandrum

Based on transcriptome sequencing (RNA-Seq) data, the expression pattern of 218 *AP2/ERF* genes in *M. dodecandrum* is shown in Figure 9 (AP2, RAV, Soloist subfamily), Figure 10 (DREB subfamily), and Figure 11 (ERF family). Forty-six genes were expressed in all 31 samples of organs and tissues. Only *MedERF-22* and *MedERF-96* were found to be highly expressed in all samples. At the same time, the *MedERF-51* gene was not detected at all. The *MdAP2/ERF* genes showed certain tissue specificity, among which 37 genes had high expression in roots, 27 in leaves, 11 in stems, nine in flower buds, and only three in fruits. Interestingly, some genes were only highly expressed in specific organs or tissues. For example, *MedDREB-41*, *MedDREB-46*, *MedDREB-68*, *MedDREB-40*, *MedDREB-49*, *MedDREB-21*, *MedDREB-16*, *MedDREB-45*, *MedDREB-58*, *MedDREB-2*, *MedDREB-78* were highly expressed in roots. *MedERF-105* and *MedERF-21* were highly expressed in flower buds. *MedERF-62* and *MedDREB-20* were highly expressed in stems. In addition, the expression of some genes in different developmental stages of the same organ had significant differences. For example, the expression of long/short stamens of *MedERF-21* and *MedERF-105* in the bud stage was significantly higher than that in the bud stage and the full flowering stage. Moreover, the expression of *MedERF-6* gene in the long/short stamens at the full flowering stage was significantly higher than that in the long/short stamens in the bud stage.

### 2.10. Response Profiles of MedAP2/ERF Genes with Hormone Treatments

Studies have shown that *AP2/ERF* family genes, as auxin-responsive genes, can affect plant growth and development under hormone-regulated expression [36]. In this study, seven genes with the highest expression level in leaves were selected as the target genes, which were most likely to respond to the regulation of auxin expression in leaves. The expression patterns of these seven *MedAP2/ERF* genes after IAA (Indole-3-acetic acid) and NPA (N-1-naphthylphthalamic acid) treatment were analyzed using RT-qPCR. The results of IAA treatment (Figure 12A) showed that the response of *MedRAV-2* gene to IAA was the most rapid, and the expression level was significantly up-regulated at 6 h after treatment, which then continuously decreased up to 24 h. The expression levels of *MedDREB13* and *MedDREB-38* increased slightly after treatment, increased to the maximum at 12 h, and returned to the pre-treatment level after 24 h. *MedDREB43* showed the same trend except that the short-term expression level did not increase significantly after treatment. The expression of *MedERF40* fluctuated after treatment, but there was no significant up-regulation or inhibition. The expression of *MedDREB7* and *MedERF31* was inhibited after treatment. However, the expression of *MedDREB38* was significantly enhanced after auxin treatment, and *MedERF31* was strongly inhibited, indicating that these two genes might be involved in the response of *M. dodecandrum* to auxin. The results of NPA treatment (Figure 12B) showed that the expression levels of *MedRAV2*, *MedDREB7*, *MedDREB13,* and *MedDREB38* increased first and then returned to the normal level. However, *MedERF40* and *MedDREB43* were continuously inhibited after NPA treatment. Unlike other genes, *MedERF31* had no significant change in gene expression in the short term after treatment, but the expression level was significantly increased after 24 h.

## 3. Discussion

### 3.1. Subfamily Member Changes for M. dodecandrum AP2/ERFs

The AP2/ERF gene family, containing at least one AP2 domain, is one of the largest families in plants and plays vital roles in plant growth, development, and stress responses [37]. Its importance in plants has garnered substantial research attention, and AP2/ERF family genes have been identified across numerous plant species (Figure 3). In *Prunus mume*, 116 AP2/ERF members were identified [38], with 271 in *Triticum durum* [39]. In this study, we uncovered a total of 218 *AP2/ERF* genes in *M. dodecandrum*. Among them, the ERF subfamily contained the highest number of members, similar to ERF members in *E. songoricum* [40]. The DREB subfamily was the most abundant in *P. patens* [10]. The number of distinct subfamily members varies across plant species, potentially reflecting preservation of suitable genes during evolution. The Ka/Ks ratios of 179 gene pairs were all less than one, indicating that strong purifying selection played an important role in the evolution of *AP2/ERF* and might largely maintained the function of *MedAP2/ERF*, which was the same as that for *Triticum turgidum* and *Eremosparton songoricum*.

### 3.2. AP2/ERF Have a Diversity Expression and by Response to IAA Function in the Tissue Development

The expression of these genes changed significantly in the tissue and developmental stages, implying the differential expression of these genes in *M. dodecandrum*. This is similar to the differential expression of AP2/ERF genes in some other species previously reported [41,42]. In *M. dodecandrum*, most of the genes specifically expressed in tissues are members of the ERF and DREB subfamilies, indicating that these genes may play an important role in regulating the development and integration of multiple organs and tissues.

It has been reported that *AP2/ERF* superfamily TFs regulate various processes of plant development and play an important role in hormone regulation and stress response [2]. According to the results of the qRT-PCR, *MedRAV2*, *MedDREB13,* and *MedDREB38* were positively regulated by auxin and auxin inhibitors, indicating that these genes might play crucial role in the response of *M. dodecandrum* to exogenous hormones, which is in accordance with the findings for tomatoes [43]. The expression of *MedERF40* and *MedERF43* increased after IAA treatment, but decreased after NPA treatment. It is speculated that it may be related to the synthesis of auxin. The expression of *MedREB7* and *MedERF31* was inhibited to varying degrees after auxin treatment, and the expression was significantly enhanced at different time periods after the treatment of auxin transport blockers, indicating that *MedREB7* may be involved in the control of *M. dodecandrum*.

As a potential fruit crop, *M. dodecandrum* presents an innovative opportunity for the production of nutritious goods and the optimization of underutilized land. Researchers and developers in agriculture have consistently strived for crops that exhibit high productivity, resilience, quality, and sustainability. However, obstacles such as limited water availability or excessive water supply, cold climates, droughts, pests, and diseases hinder advancements in plant yield per unit area. *AP2/ERF* is a prominent TF superfamily with crucial roles in plant reproductive growth and vegetative growth as well as stress-induced responses. In this study, we provide an overview of the categorization, characteristics, functionality, regulatory mechanisms, and potential uses of *AP2/ERF* transcription factors in *M. dodecandrum*. Despite the identification of certain *AP2/ERF* regulators and their molecular functions, there is still a lack of comprehensive studies. In reality, the investigation of *AP2/ERF* transcription factors in this recently cultivated plant is currently in its preliminary phase and the examination of its underlying mechanisms is being conducted within a laboratory setting rather than out in the field. One future challenge involves investigating the upstream and downstream components of *AP2/ERF* TFs, as well as exploring potential connections between regulatory pathways in field conditions. Another challenge lies in finding a solution to the trade-off effect observed between achieving high yield and resistance while maintaining environmentally friendly, organic production practices. Leveraging our understanding of the molecular mechanisms governing these traits, breeders can utilize beneficial alleles controlling these characteristics to develop desired *M. dodecandrum* cultivars that exhibit both high yield and disease resistance, alongside sustainable, eco-friendly production methods.

## 4. Materials and Methods

### 4.1. Identification and Physicochemical Properties of AP2/ERF Genes in M. dodecandrum Genome

The chromosome-level *M. dodecandrum* genome data provided a retrieval template in this study. The Hidden Markov model (HMM) with PF00847 (AP2 domain) was used to match the *AP2/ERF* gene sequences in *M. dodecandrum* though The TBtools-Ⅱ v2.019 [44] software and we performed domain checking to remove sequences without the AP2 domain. ProtParam (https://web.expasy.org/protparam/, accessed on 16 August 2023) online analytical tools were used to predict the number of amino acid of *M. dodecandrum* AP2/ERF proteins, molecular weight, theoretical pI, instability index, the aliphatic index, and the grand average of hydropathicity.

### 4.2. Analysis of AP2/ERF Gene Structure, Conserved Domains and Motifs in M. dodecandrum

Online software NCBI BatchCD-search (https://www.ncbi.nlm.nih.gov/Structure/bwrpsb/bwrpsb.cgi, accessed on 16 August 2023) and MEME (https://meme-suite.org/meme/doc/, accessed on 16 August 2023) were used to analyze the conserved domains. Online tools were employed to analyze the motifs of 218 genes in the *AP2/ERF* gene family of *M. dodecandrum*, with the number of motifs set to 10 and other parameters defaulted. Finally, the gene structure, conserved structural domains, and conserved motifs were visualized using TBtools software.

### 4.3. Prediction of Secondary and Tertiary Structure of AP2/ERF Transcription Factor Proteins

In this study, proteins (MedAP2-17, MedDREB-38, MedERF-92, MedRAV-1, and Medsoloist-3) with minimal branches of each type were selected according to the phylogenetic tree of *AP2/ERF* proteins, which were conserved and represented the structural characteristics of the entire subfamily. We used online tools, SOPMA (https://npsaprabi.ibcp.fr, accessed on 16 August 2023), online websites, and SWIS-MODEL (https://swissmodel.expasy.org/, accessed on 16 August 2023) to predict the secondary structures and tertiary structures of the MedAP2/ERF members.

### 4.4. Chromosome Distribution and Promoter Capture of the AP2/ERF Genes

The *MedAP2/ERF* gene was mapped using TBtools-Ⅱ software, and the synteny relationships between the genes were analyzed and visualized.

Gene duplications were also studied based on the Plant Genome Duplication Database. For segmental duplications, a BLASTP search was performed against all identified peptide sequences of identified *MedAP2/ERF* in common bean. As potential anchors, the top five matches with an e-value ≤ 1 × 10^−5^ were evaluated and then MCScan was utilized for the determination of collinear blocks. Finally, the alignment was performed and e-value ≤ 1 × 10^−10^ was selected as a significant match. 

The cis-element site analysis was performed for the 2000 bp region upstream of the transcription start site of 218 *MedAP2/ERF* genes. Extraction was performed using TBtools. The PlantCARE website was used to analyze the cis-element sites and data were finally visualized using Microsoft Excel 2019 and Origin.

### 4.5. Phylogenetic Analyses against Plant APG Ⅳ System and Selection Pressure of AP2/ERF Family in M. dodecandrum

We downloaded 176 *Arabidopsis AP2/ERF* predicted amino acid sequences from the Plant Transcription Factor Database (PlantTFDB V5.0). The protein sequences of 218 *MedAP2/ERF*s from *M. dodecandrum* and 176 *AtAP2/ERFs* from *A. thaliana* were used to construct basic phylogenetic trees. Multiple sequence alignment was performed using PhyloSuite v1.2.2 software [45]. Phylogenetic trees were constructed by the maximum likelihood method (with 1000 bootstrap replicates) using the IQtree from PhyloSuite. The APG IV system locations of the 24 species were constructed using NCBI CommonTree (https://www.ncbi.nlm.nih.gov/taxonomy/CommonTree, accessed on 16 September 2023). Subfamily taxonomic and genetic data of 24 species were obtained from previous studies [40]. The phylogenetic trees were visualized using the iTOL website (https://itol.embl.de/, accessed on 16 September 2023) and Excel2021.

The simple Ka/Ks calculator function in TBtools software was used to calculate the non-synonymous substitution rate (Ka) and synonymous substitution rate (Ks) of repeated gene pairs in the AP2/ERF gene family of *M. dodecandrum*. The Ka/Ks ratio of the gene was obtained, and the gene pairs that did not meet the threshold were removed for selection pressure analysis.

### 4.6. Micro RNA Editing for the AP2/ERF Family Members

The detection of miRNA target genes has played a pivotal role in elucidating the functional mechanisms of miRNAs. In this study, we employed bioinformatics and prediction analyses using the web-based psRNA Target Server (https://www.zhaolab.org/psRNATarget/analysis, accessed on 19 September 2023) to investigate miRNAs and their target *MedAP2-ERF* genes. The expected value was set to 5, and the remaining parameters were defaulted. Alignment of identified genes with the miRNAs of *A. thaliana* was conducted and labeled.

### 4.7. Analysis of Gene Expression Patterns

To investigate the potential involvement of *AP2/ERF* genes in various organs of *M. dodecandrum*, we conducted a comprehensive analysis of the expression patterns of 218 *AP2/ERF* genes across roots, stems, leaves, and flower buds (including whole buds and specific components such as long/short stamens, pistils, sepals at different developmental stages), as well as fruits (encompassing expansion period, coloring period and full-ripening period). The TBtools software was utilized for data visualization.

### 4.8. RT-qPCR Assay during the IAA Induction

In this experiment, the *M. dodecandrum* planted in the Soil and Water Conservation Park of Fujian Agriculture and Forestry University (E 119°13′48″, N 26°6′36″) was used as the experimental material. To ensure the consistency of the experiment and the stability and repeatability of results, independent plants were selected from the nursery. Five individuals as bio-duplications for each treatment were prepared. Untreated plants with the same growth status were used as a control. Before the experiment, *M. dodecandrum* was transplanted into a nutrient–soil matrix and grown under natural conditions (24–36 °C, 16 h of light/8 h of darkness) for ten days. The whole plant was sprayed with hormones or inhibitors and tender leaves at the top of plants were collected at corresponding time points. The treatment methods were exogenous auxin, 3-Indoleacetic acid (IAA, 100 μm), auxin transport inhibitor, and *N*-1-Naphthylphthalamic acid (NPA, 100 μm) solution. The plant surface was sprayed with an atomizer until it was fully moist, but no droplets condensed. The leaves of *M. dodecandrum* were sampled at 6, 12, 24 h after IAA treatment and 6 and 24 h after NPA treatment. The samples were loaded into a 2.5 mL sterile non-enzymatic cryopreservation tube, which was quickly frozen in liquid nitrogen. Finally, the total RNA of *M. dodecandrum* was extracted according to the method of R6827 Plant RNA Kit (Omega Bio-Tek, Guanghzhou, China). In this step, all the samples were digested by DNase to eliminate DNA from total RNA extracts. The Hieff UNICON ^®^ Universal Blue qPCR SYBR Green Master Mix kit (Yeasen Biotechnology, Shanghai, China) was used to synthesize quantitative cDNA single strand by reverse transcription of 2 mg RNA. The cDNA was used as a template for real-time quantitative PCR detection. The *MedActin1* gene was selected as the internal reference by fluorescence quantification. The sequences of the genes used in the reaction and the internal reference primers are shown in Appendix A. The reaction was designed with three technical replicates. The reaction system was 20 μL, which included Hieff UNICON^®^ Universal Blue qPCR SYBR Green Master Mix 10 μL, Forward Primer (10 μM) 0.5 μL, reverse primer (10 μM) μL, template DNA 2 μL, and sterile ultrapure water 7 μL. The amplification program was 95 °C pre-denaturation for 30 s, 95 °C for 4 s, 60 °C for 34 s, and 95 °C for 15 s, a total of 40 cycles. The resultant data were calculated using the 2^−ΔΔCt^ method to calculate the relative gene expression levels and were processed using one-way analysis of variance (ANOVA). *t*-tests were performed based on GraphPad Prism 9 to compare the statistical differences, with *p* < 0.05 as * and *p* < 0.01 as **, and finally, visualization and analysis were performed using origin.

## 5. Conclusions

In this study, we identified 218 genes and classified them into five main subfamilies of *M. dodecandrum*. Through comparative analysis of the phylogenetic relationships among these genes, we determined that the diversification and conservation of the AP2/ERF family were primarily the result of segmental duplication, contributing to the expansion of the super gene family. Paralogous genes within a group or subgroup may possess redundant functions. The gene structure, cis-acting elements, and miRNA splicing sites revealed multiple hormone responses and abiotic stresses through transcriptional and post-transcriptional regulations. Our findings demonstrated a significant involvement of *MedAP2/ERFs* in tissue formation and their response to developmental modifications induced by the IAA hormone. The findings will also provide valuable insights into the role of *MedAP2/ERF*s genes in governing agronomic, economic, and ecological traits in *M. dodecandrum*, and potentially in other beneficial plant species.

## Figures and Tables

**Figure 1 ijms-24-16362-f001:**
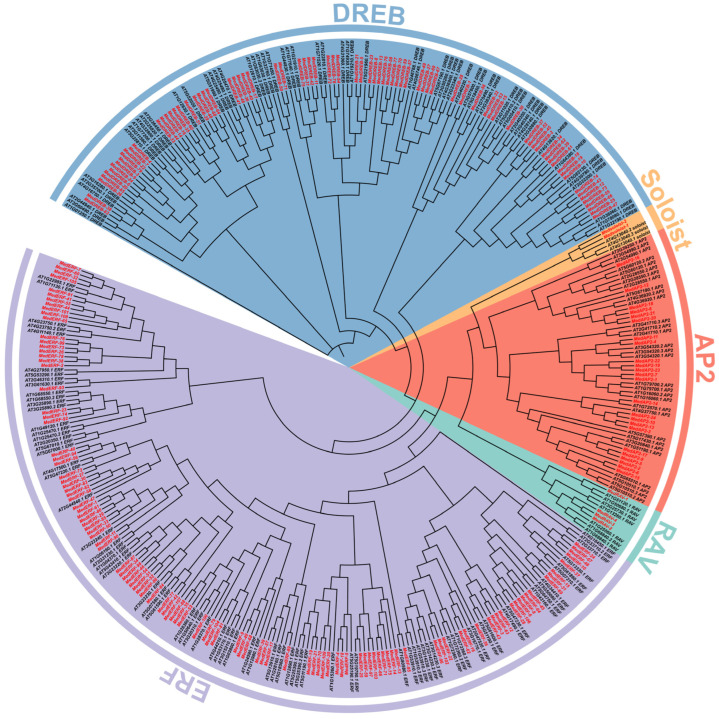
Phylogenetic analysis of AP2/ERF genes in *Melastoma dodecandrum*. The ML (maximum likelihood) phylogenetic tree was constructed using PhyloSuite with 1000 bootstraps. AP2, DREB, ERF, RAV, Soloist subfamilies are grouped together as indicated by red, blue, purple, green, and yellow, respectively. *MedAP2/ERFs* are indicated by red text, and *AtAP2/ERF*s are indicated by black text.

**Figure 2 ijms-24-16362-f002:**
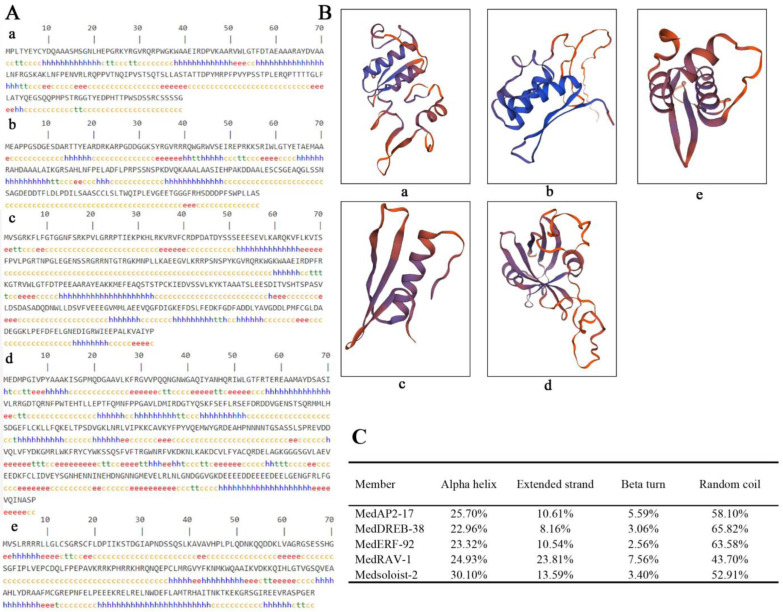
The prediction of AP2/ERF subfamily structures (**A**). Secondary structure of each AP2/ERF subfamily in *M.dodecandrum*. (**B**) Three-dimensional structures of typical AP2/ERF subfamily in *M.dodecandrum*. Note: (**a**) MedAP2-17 in AP2; (**b**) MedDREB-38 in DREB; (**c**) MedERF-92 in ERF; (**d**) MedRAV-1 in RAV; (**e**) Medsoloist-2 in soloist. The odd-numbered rows represent amino acid sequences, and the even-numbered rows represent their corresponding secondary structure types. Alpha helix is shown in blue; green is beta turn; yellow indicates random coil; red is extended strand. (**C**). Secondary structure ratio of MedAP2/ERF protein.

**Figure 3 ijms-24-16362-f003:**
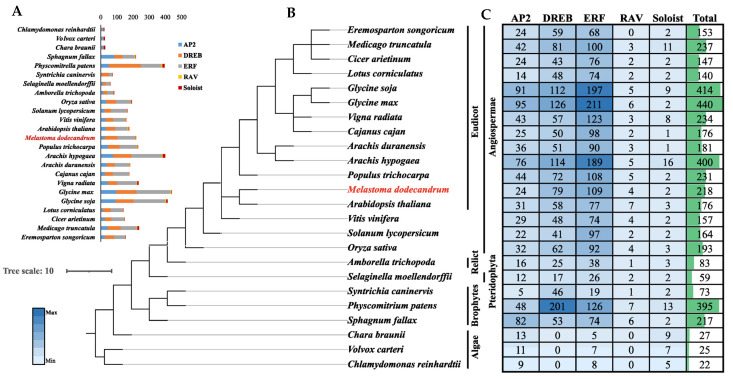
APG IV phylogenetic positions, and classification and number of AP2/ERF gene family members across 24 plant species. (**A**) The barplot of AP2/ERF superfamily numbers. (**B**) The APG IV phylogenetic positions of uesd 24 species. (**C**) The number of gene members in each species is displayed as a heatmap (color bar) and bar chart, with deep blue indicating the maximum value and light blue denoting the minimum (0). The total number of AP2/ERF family members in the 24 plant species is shown by the green bar chart.

**Figure 4 ijms-24-16362-f004:**
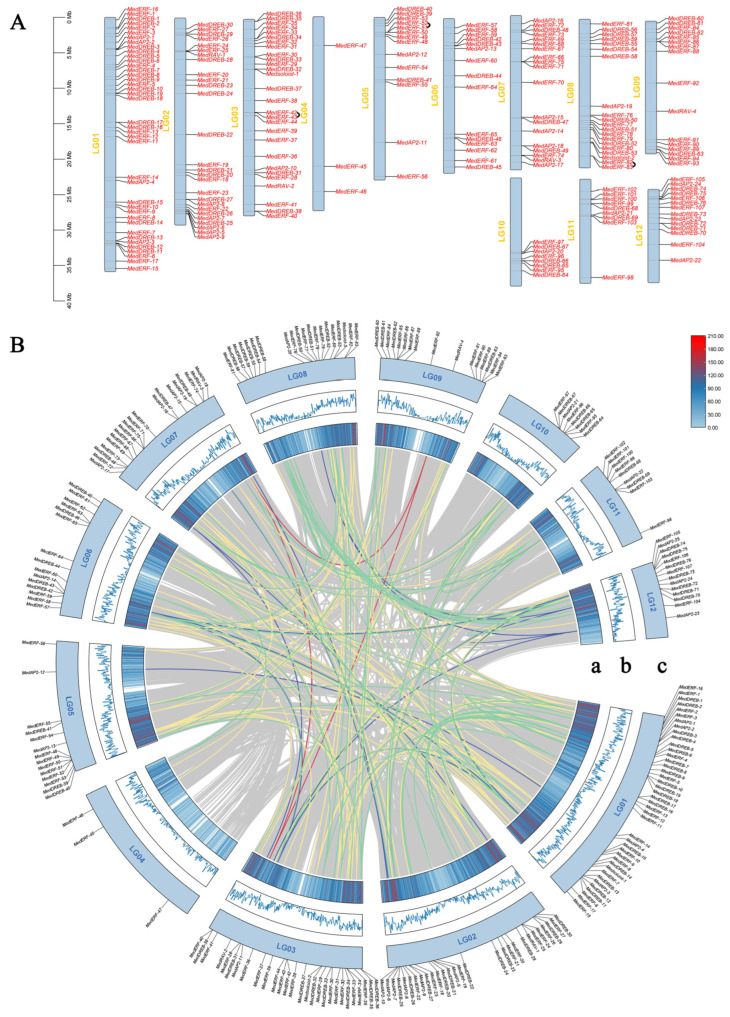
Chromosome localization and synteny analysis of *MedAP2/ERF* genes. (**A**) Chromosome localization of *MedAP2/ERF* gene. The chromosome numbers are marked on the left side of each chromosome. The right side of each chromosome is marked with gene names. The black lines link genes that have tandem repeats. (**B**) Synteny analysis of *MedAP2/ERF* gene. Overlapping genes are connected by line segments, with different colored lines linking different gene families. The blue line links the AP2 subfamily; the green line links the REB subfamily; the yellow lines link the ERF subfamily; the red line links the RAV subfamily; the light blue line links the soloist subfamily. (**a**) Line colors indicate gene density. (**b**) The blue line represents gene density. (**c**) LG01–LG12 is representative of *M. dodecandrum* chromosome.

**Figure 5 ijms-24-16362-f005:**
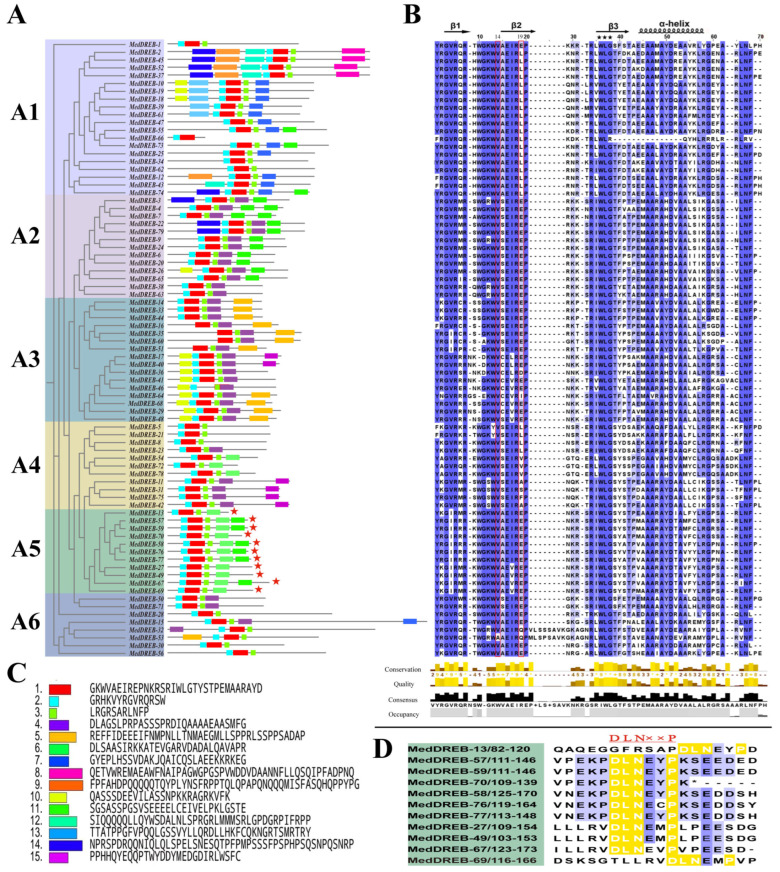
Classification and sequence analysis of MedDREBs subfamily. (**A**) Phylogenetic tree and motif distributions in A1–A6 group of MedDREB subfamily. Genes with EAR motif are indicated by red stars. (**B**) Multiple sequence alignment of AP2 domains in MedDREBs. Genes with WLG conserved motifs are indicated by black stars. (**C**) Amino acid sequence of each motif. (**D**) MedDREB genes with EAR motif.

**Figure 6 ijms-24-16362-f006:**
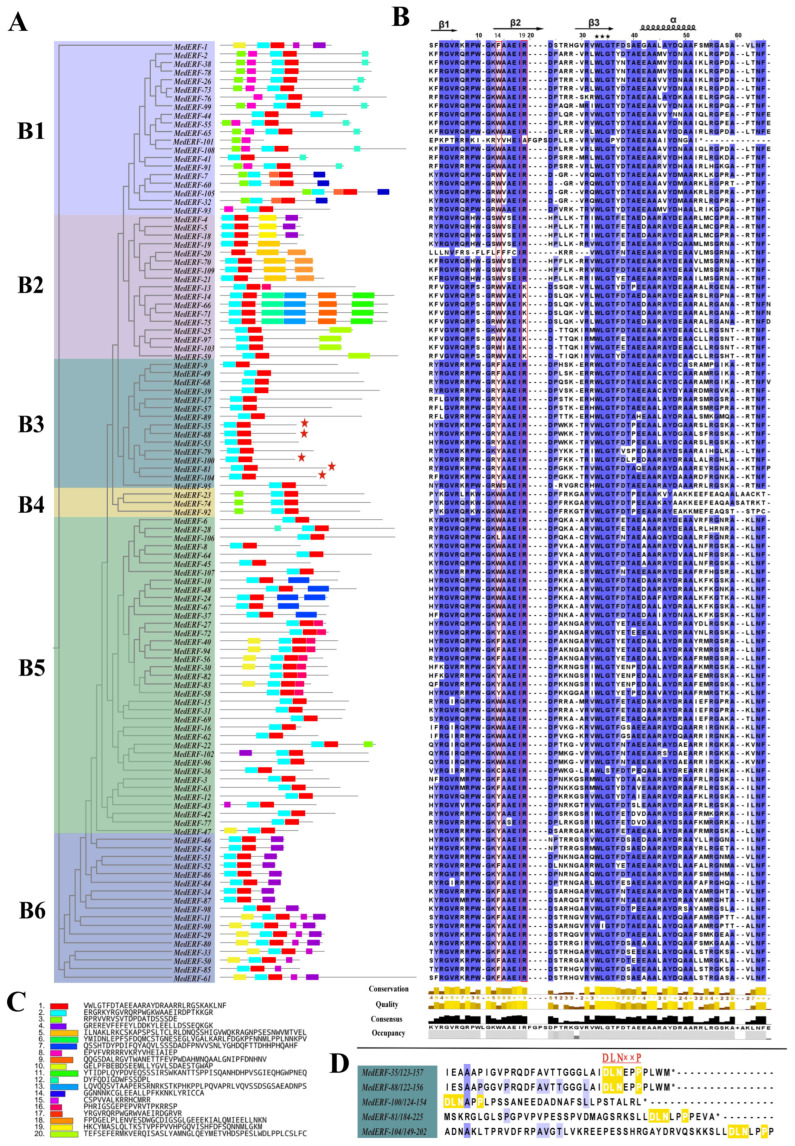
Classification and sequence analysis of *MedERF* subfamily. (**A**) Phylogenetic tree and motif distributions in B1–B6 group of *MedERF* subfamily. Genes with EAR motif are indicated by red stars. (**B**) Multiple sequence alignment of AP2 domains of *MedERF*s. Genes with WLG conserved motifs are indicated by black stars. (**C**) Amino acid sequence of each motif. (**D**) *MedERF* genes with EAR motif.

**Figure 7 ijms-24-16362-f007:**
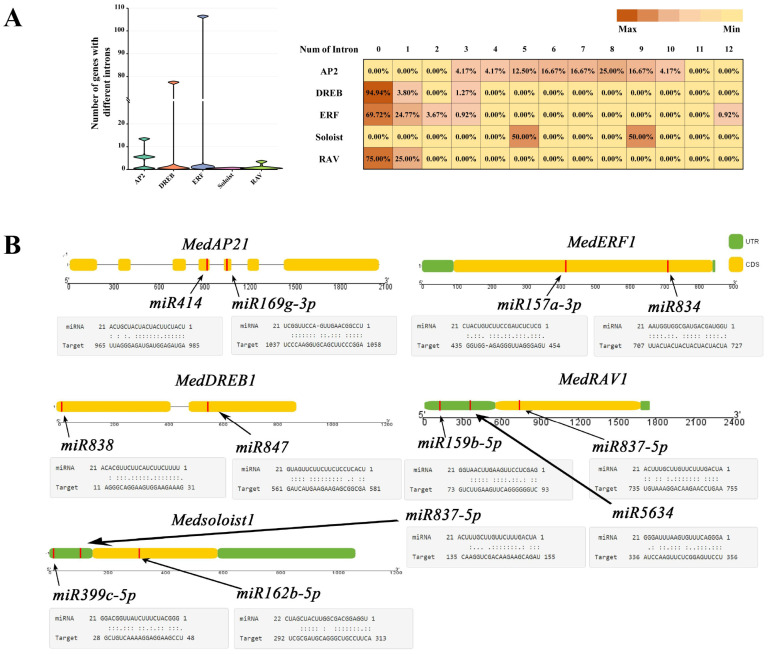
The introns and miRNA splicing site in *MedAP2/ERF* gene. (**A**) Statistics for *AP2/ERF* gene intron–exon number in *M. dodecandrum.* (**B**) Targets for some microRNAs. This view displays partial regulatory roles of miRNAs and AP2/ERF transcription factors. Yellow parts show the coding region of AP2/ERFs. Red lines marked the splicing sites.

**Figure 8 ijms-24-16362-f008:**
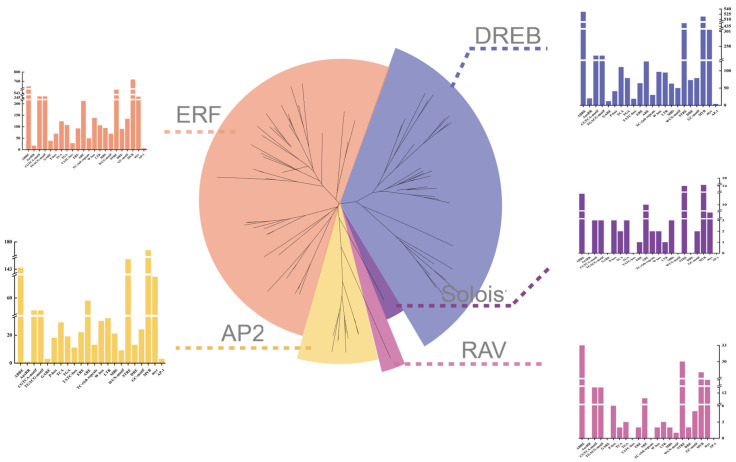
Cis-elements in the putative promoter regions of *MedAP2/ERF* genes. The center phylogenetic tree presented the classification of *AP2/ERF* subfamily. The numbers of different cis-elements in the *MedAP2/ERF* genes are indicated by numbers and different colors in the histogram.

**Figure 9 ijms-24-16362-f009:**
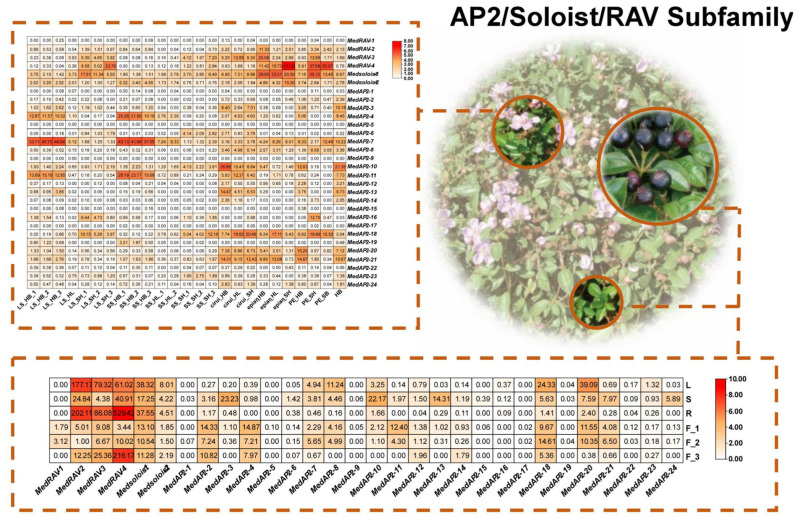
Heat map showing the expression levels of AP2, Soloist, and RAV subfamily genes in five organs. The expression of all genes identified in this study was determined using transcriptome sequencing (RNA-Seq) analysis of *M. dodecandrum*. LS_HB: long stamens of big bud; LS_HL: long stamens of small bud; LS_SH: long stamens of blooming stage; SS_HB: short stamens of big bud; SS_HL: short stamens of small bud; SS_SH: short stamens of blooming stage; cirui_HB: pistils of big bud; cirui_HL: pistils of small bud; cirui_SH: pistils of blooming stage; epian_HB: sepals of big bud; epian_HL: sepals of small bud; epian_SH: sepals of blooming stage; PE_HB: petals of big bud; PE_SH: petals of blooming stage; PE_SB: petals in decay period; L: Leaf; R: Root; F: Fruit; HB: Whole bud; S: Stem. The color scale indicates the normalized log 2 conversion count per million kilobases of reading, where white indicates low levels and orange indicates high levels.

**Figure 10 ijms-24-16362-f010:**
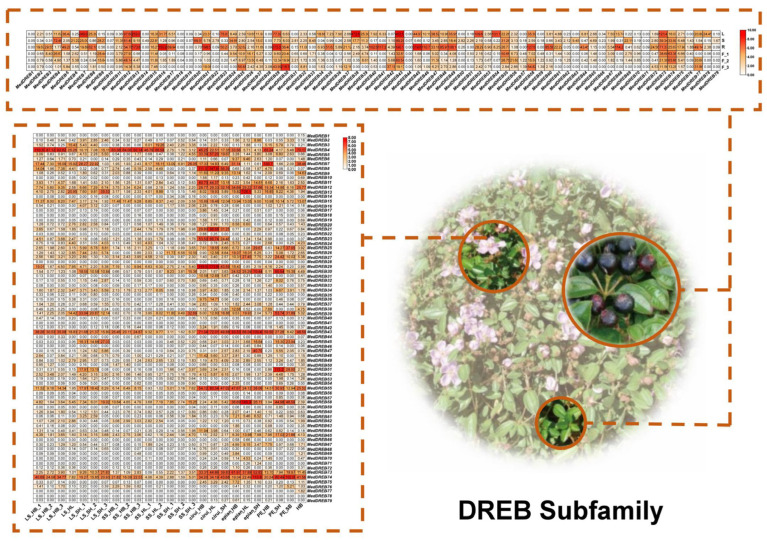
Heat map showing expression levels of the DREB subfamily gene in five organs.

**Figure 11 ijms-24-16362-f011:**
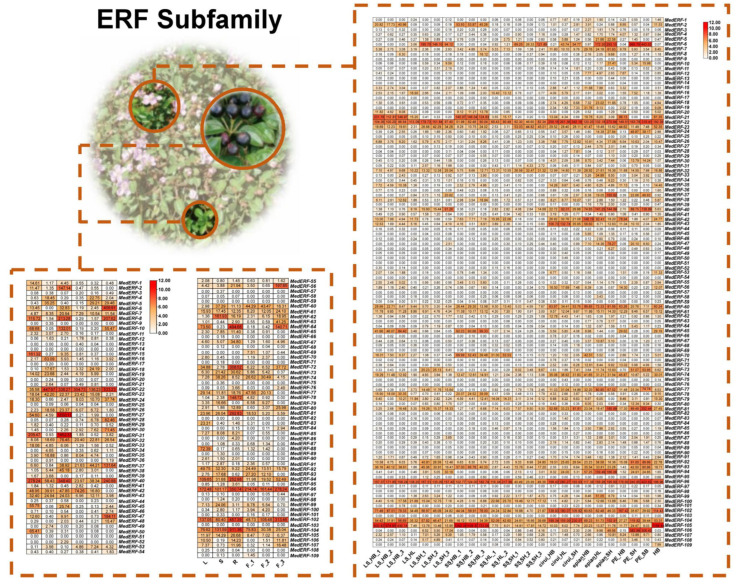
Heat map showing expression levels of the ERF subfamily gene in five organs.

**Figure 12 ijms-24-16362-f012:**
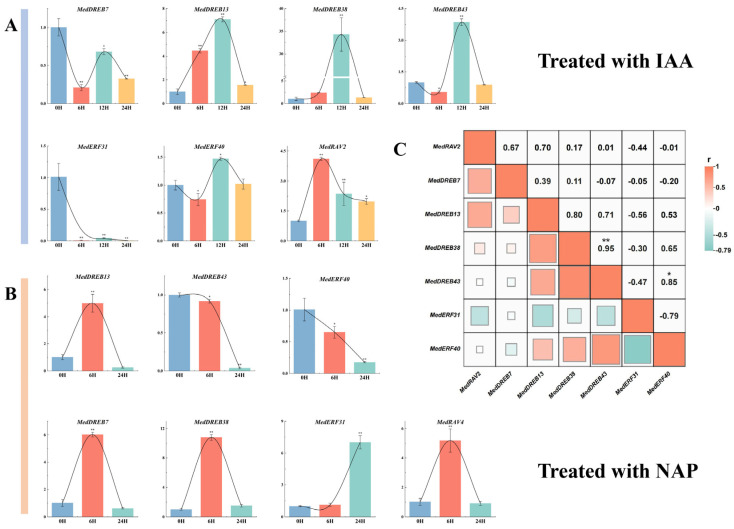
*MedAP2/ERF* gene expression at 0 h, 6 h, 12 h and 24 h after 100 μmol/L IAA treatment and at 0 h, 6 h, and 24 h after 100 μmol/L NPA treatment. (**A**) The expression of MedAP2/ERF genes with IAA treatment; (**B**) The expression of MedAP2/ERF genes with NPA treatment. Relative transcript levels are calculated by real-time PCR with MedActin1 as a standard. Data are means ± SE of three separate measurements based on *t*-test, taking *p* < 0.05 as * and *p* < 0.01 as **. (**C**) Heatmap shows the correlation of gene expressions between the two treatments.

## Data Availability

The original genome sequences described in this article have been submitted to National Genomics Data Center (NGDC, https://ngdc.cncb.ac.cn, accessed on 16 August 2023) under accession number PRJCA005299; raw transcriptome data were stored at GSA, portion accession number: CRA004347. All data generated or analyzed during this study are included in this published article or Appendix A and are also available from the corresponding author on reasonable request. Comparative data were obtained from the Plant Transcription Factor Database 5.0 website (http://planttfdb.gao-lab.org/, accessed on 16 August 2023).

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
