# Peer review of "Bioinformatic Assessment and Expression Profiles of the AP2/ERF Superfamily in the Melastoma dodecandrum Genome"

_ijms, 2023, doi:10.3390/ijms242216362_

Round 1

Reviewer 1 Report

Comments and Suggestions for Authors

In this paper, the authors presented original data on the sequencing of AP2/ERF family genes for M. dodecandrum. The fruits of this plant contain significant amounts of tannins and flavonoids with high antioxidant activity. M. dodecandrum is considered a valuable raw material with significant medicinal value.

In their study, the authors identified AP2/ERF members in Melastoma dodecandrum based on genome data and, based on phylogenetic analysis, classified them into relative subfamilies connecting to the AP2/ERF family in 20 other representative plant species. They detailed analysis of basic physical and chemical properties, gene structure, conserved motif, chromosome arrangement, cis-element binding sites, and microRNAs. They located potential core functional elements based on gene expression patterns in different tissues. The study is a pioneering approach in the genome-wide identification of M. dodecandrum AP2/ERF family genes, providing valuable information on the response and regulatory mechanisms of AP2/ERF genes in this newly domesticated plant. Moreover, they proposed potential candidate genes for future plant breeding endeavors.

As a fruit plant, M. dodecandrum offers innovative opportunities for the production of nutritional commodities and the optimization of underutilized land.

The work contains original, very well-prepared data. The graphical presentation of data deserves special recognition. I recommended that this MS be published. 

Author Response

R1: Thank you for your valuable suggestions, we added more details about such as how many plants were used in the experiments, how many times the experiment was repeated, what are the control plants, what kind of samples were collected, the number and quantity of samples, total amount of RNA used for qRT-PCR, how DNA is eliminated from total RNA extracts. The changing part are listed as follow:

Line523-557 ”Five individuals as bio-duplications for each treatment were prepared. Untreated plants with the same growth status were used as control. Before the experiment, M. dodecandrum was transplanted into a nutrient soil matrix and grown under natural conditions (24-36 °C, 16 hours of light / 8 hours of darkness) for ten days. The whole plant was sprayed with hormones or inhibitors and tender leaves at the top of plants were collected at corresponding time points. ”

R2: It is a pity that we can not supply the Northern blot experiment in this paper because of lacking instruments. However the results are still believable, we used multiple biological replicates to reduce the error, the results presented in the bar chart also show smaller error lines. This shows that in this experiment, the corresponding gene was indeed induced.

R3: We changed the figures as your request, this really help display our results.

Minor:

L88-89: the genome data uesd in the MS were listed in the end of MS (Data availability part):

”The original genome sequences described in this article have been submitted to National Genomics Data Center (NGDC, https://ngdc. cncb.ac.cn) under accession number PRJCA005299, raw transcriptome data were stored at GSA, portion accession number: CRA004347.”

We have perfromed an overall inspection, especially targeted at the problem you rise in L23, L163, L165, and a native English speaker helped us revised the final manuscript,

As for L90 and 92, we changed and added necessary link to the software.

Reviewer 2 Report

Comments and Suggestions for Authors

Reviewing the manuscript “Bioinformatic assessment and expression profiles of AP2/ERF superfamily in Melastoma dodecandrum genome submitted by Zhou et al. in the International Journal of Molecular Sciences was a pleasure. Though the manuscript is well organized, it requires extensive language correction and a couple of experiments.

Major:

Section 2.8. RT-qPCR assay during the IAA induction: More details are required in this section, such as how many plants were used in the experiments, how many times the experiment was repeated, what are the control plants, what kind of samples were collected, the number and quantity of samples, total amount of RNA used for qRT-PCR, how DNA is eliminated from total RNA extracts.

Gene expression studies are very tricky in qPCR assay. Hence, a complementary assay such as Northern blot is highly recommended.

Section 3.7. Target of specific miRNAs for MedAP2/ERF genes: A wet lab experiment on at least two targets is required to prove the role of predicted miRNAs in regulating predicted targets.

Minor:

L23: “…induced by IAA hormones…”. Expand IAA as this term is introduced. Similarly, it is better to introduce terms such as AP2, ERF, and so on throughout the text when it is used for the first time. 

L88-89: “The chromosome-level M. dodecandrum genome data provided retrieval template in this study.” Please provide the link and Accession number from where the data was retrieved.

L90: ”… though The TBtools[36] software…” change to ”… though the TBtools[36] software…

L91-92: provide a link to the Prot-Param online tool.

L163: “100um), auxin” change to “100µm), auxin”. Please make similar changes elsewhere in the manuscript.

L165: “…were sampled at 6,12,24..”. Leave a space after the comma. Make similar changes throughout the manuscript.

Comments on the Quality of English Language

A language correction by a native English speaker who is a biology graduate is required

Author Response

(The authors gave the same response as above.)

Round 2

Reviewer 2 Report

Comments and Suggestions for Authors

None